# Evidence for non-steady-state carbon emissions from snow-scoured alpine tundra

John F. Knowles [1,2], Peter D. Blanken[3], Corey R. Lawrence [4] & Mark W. Williams[1,3]

High-latitude warming is capable of accelerating permafrost degradation and the decomposition of previously frozen carbon. The existence of an analogous high-altitude feedback, however, has yet to be directly evaluated. We address this knowledge gap by coupling a radiocarbon-based model to 7 years (2008–2014) of continuous eddy covariance data from a snow-scoured alpine tundra meadow in Colorado, USA, where solifluction lobes are associated with discontinuous permafrost. On average, the ecosystem was a net annual source of $232 \pm 54$ g C m$^{-2}$ (mean ± 1 standard deviation) to the atmosphere, and respiration of relatively radiocarbon-depleted (i.e., older) substrate contributes to carbon emissions during the winter. Given that alpine soils with permafrost occupy $3.6 \times 10^6$ km$^2$ land area and are estimated to contain 66.3 Pg of soil organic carbon (4.5% of the global pool), this scenario has global implications for the mountain carbon balance and corresponding resource allocation to lower elevations.

[1] Institute of Arctic and Alpine Research, University of Colorado Boulder, UCB 450, Boulder, CO 80309-0450, USA. [2] School of Geography and Development, The University of Arizona, Earth and Natural Resources 2 Building, South 4th Floor, Tucson, AZ 85721-0137, USA. [3] Department of Geography, University of Colorado Boulder, UCB 260, Boulder, CO 80309-0260, USA. [4] U.S. Geological Survey, P.O. Box 25046, Denver Federal Center, MS 980, Denver, CO 80225, USA. Correspondence and requests for materials should be addressed to J.F.K. (email: Johnknowles@email.arizona.edu)

Heterotrophic respiration of thawed permafrost carbon is considered one of the most likely positive climate feedbacks between high-latitude ecosystems and the atmosphere[1,2]. Yet, the magnitude of this feedback represents a highly uncertain component of global climate change predictions[3,4], and the potential contribution of high-altitude permafrost systems to this phenomenon remains virtually unknown[5,6]. Recent work, however, has shown that a variety of alpine shrubland and meadow ecosystems are net annual sources of carbon to the atmosphere[7–10], and there is evidence of permafrost degradation in mountain ranges around the world that function as indicator systems and source areas of water and nutrients to lower elevations[11–14]. Since lower temperature soils are more sensitive to warming[15,16], climate change could have a disproportionate impact on high-altitude soil organic matter (SOM) reserves, but ecosystem response to permafrost degradation is complicated by changes in plant-soil interactions, hydrology, and nutrient fluxes that operate on dissimilar timescales[10,17–20].

High-latitude tundra ecosystems are currently net annual carbon dioxide ($CO_2$) sources to the atmosphere, but large standing carbon stocks indicate periods of carbon sequestration in the past[21]. This carbon sink-to-source threshold coincides with a period of recent rapid climate warming, suggesting that the direct and/or indirect effects of permafrost degradation are capable of accelerating decomposition to the degree that it exceeds vegetation primary productivity[21,22]. Measurement of soil radiocarbon is a powerful tool with which to investigate temporal carbon cycling dynamics because it can be used to approximate the overall mean decay rate of carbon in soils[23,24]. In particular, when compared across soil carbon pools of differing mean age isolated by density separations or other methods, the radiocarbon signature of soil respiration can constrain the relative source of each pool to soil-derived $CO_2$. In this study, we used a density fractionation method to partition soil into three pools corresponding to relatively fast, intermediate, and slowly cycling carbon (described in Methods section)[25]. We then used the data to parameterize a multi-pool soil carbon model in order to calculate the relative contribution of these carbon sources to the measured radiocarbon content of carbon dioxide ($^{14}CO_2$) from a long-term alpine tundra research site on Niwot Ridge, Colorado, USA.

The Niwot Ridge site is believed to be the highest-elevation eddy covariance tower in North America (40°03′07″ N; 105°35′11″ W; 3494 m above sea level)[26,27], and is one of the only sites equipped to evaluate the seasonal carbon dynamics of the potentially vulnerable alpine tundra ecosystem (Fig. 1a; see also ref. [28]). In the Rocky Mountains, snow-scoured alpine tundra results from persistently windy conditions, and these exposed locations are common above the alpine treeline[29]. On Niwot Ridge, frost creep and solifluction have produced lobes of slowly moving soil on gentle slopes with negligible snow cover[30–32]. Although these locations represent only a fraction of the heterogeneous alpine environment (Fig. 1b), they are the most active geomorphic, hydrologic, and biologic zones in the area[33], and function as control points of winter biological activity on the snow-scoured landscape[34,35]. Locations with solifluction deposits were previously believed to support permafrost[30,36], but recent work suggests that soil temperature now rises above 0 °C at both 2 and 4 m depths for several weeks in summer[33,37]. Given that the environmental conditions at these sites were marginal for the presence of permafrost in the 1970s, and that summer air temperatures and the number of positive degree days have been increasing on Niwot Ridge since that time[37,38], respiration of previously frozen substrate could contribute to currently observed carbon emissions at this site.

The constituent carbon pools and radiocarbon content of alpine tundra soils at this location have not been previously coupled to eddy covariance measurements of the net ecosystem exchange of $CO_2$ (NEE); therefore, this work represents a unique opportunity to test the degree to which the alpine tundra ecosystem may be responding to environmental-driven or disturbance-driven changes. We specifically set out to characterize the net annual carbon source strength of a high-altitude alpine tundra site and to evaluate the relative contributions of distinct soil carbon pools to the observed ecosystem respiration fluxes on a seasonal basis. Here, we show sustained net annual carbon emissions including respiration of older carbon substrate during the winter, both of which suggest non-steady-state ecosystem function. The degree to which this ecosystem parallels the increasingly well-characterized paradigm of arctic tundra carbon cycling will constrain the likelihood and magnitude of an alpine tundra permafrost carbon feedback to environmental change or disturbance.

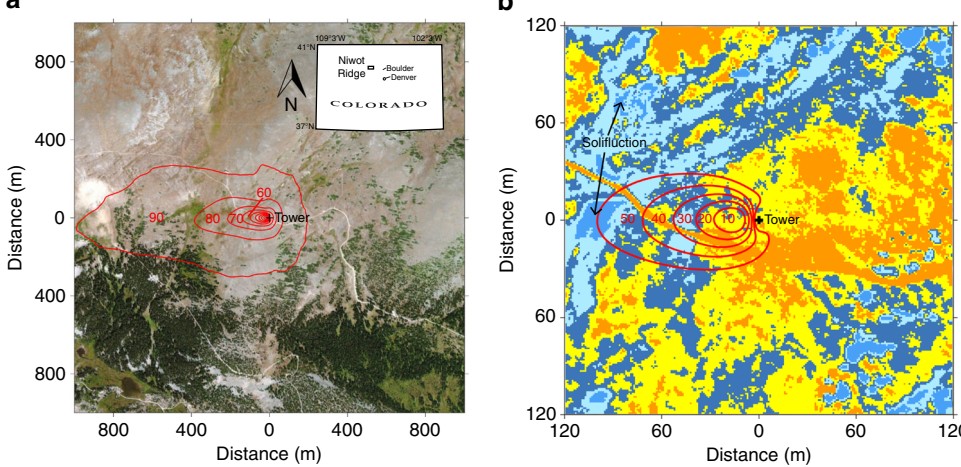

**Fig. 1** Two-dimensional representation of the flux tower footprint. Contours denote the cumulative percent contribution (in red) of heterogeneous alpine tundra source areas to the net ecosystem exchange of carbon dioxide (NEE) flux between 1 January 2008 and 31 December 2009[55]. **a** Inset shows the location of Niwot Ridge in Colorado, USA (40°03′ N; 105°35′ W). Background image ©2018 Google. **b** A k-means clustering algorithm separates the tundra into five unsupervised land cover classes based on the red-green-blue spectrum to demonstrate the scale of spatial heterogeneity in the area of peak measurement sensitivity[55]. Radiocarbon data were collected from areas of dry, moist, and wet meadow vegetation tundra including two solifluction lobes (labeled) associated with permafrost

| Table 1 Meteorological, growing season, and carbon cycling variability from 2008 through 2014 | | | | | |
|---|---|---|---|---|---|
| Year | 1 Jun–31 Aug $T_a$ (°C) | 1 Sep–31 May $T_a$ (°C) | MAAT (°C) | 1 Apr–15 Jun PPT (mm) | 16 Jun–15 Sep PPT (mm) | TAP (mm) |
| 2008 | 9.2 | −5.5 | −1.8 | 279 | 220 | 1119 |
| 2009 | 8.2 | −4.7 | −1.5 | 211 | 124 | 916 |
| 2010 | 9.5 | −4.3 | −0.8 | 268 | 128 | 1046 |
| 2011 | 10.0 | −5.3 | −1.4 | 281 | 333 | 1307 |
| 2012 | 11.0 | −3.3 | 0.3 | 86 | 275 | 846 |
| 2013 | 10.0 | −5.4 | −1.5 | 149 | 252 | 956 |
| 2014 | 8.6 | −4.4 | −1.1 | 156 | 186 | 869 |
| Mean | 9.5 | −4.7 | −1.1 | 204 | 217 | 1008 |
| SD | 0.9 | 0.8 | 0.7 | 76 | 77 | 163 |

| Year | GS start (doy) | GS end (doy) | GSL (d) | GS GPP (g C m$^{-2}$) | GS $R_{eco}$ (g C m$^{-2}$) | NEE (g C m$^{-2}$) |
|---|---|---|---|---|---|---|
| 2008 | 166 ± 4 | 241 ± 11 | 80 ± 16 | 123.8 | 68.5 | 177.4 |
| 2009 | 171 ± 4 | 238 ± 22 | 74 ± 29 | 115.0 | 60.1 | 194.1 |
| 2010 | 168 ± 4 | 232 ± 15 | 69 ± 17 | 106.3 | 53.0 | 177.1 |
| 2011 | 170 ± 18 | 242 ± 11 | 77 ± 25 | 122.0 | 66.4 | 261.5 |
| 2012 | 163 ± 12 | 242 ± 9 | 81 ± 17 | 126.5 | 72.2 | 224.1 |
| 2013 | 159 ± 8 | 241 ± 9 | 87 ± 17 | 126.1 | 73.7 | 314.0 |
| 2014 | 169 ± 15 | 233 ± 17 | 69 ± 28 | 107.8 | 53.5 | 278.7 |
| Mean | 167 ± 11 | 238 ± 13 | 77 ± 21 | 118.2 | 63.9 | 232.4 |
| SD | 4 ± 6 | 4 ± 5 | 7 ± 6 | 8.5 | 8.5 | 53.6 |

$T_a$, air temperature; MAAT, mean annual air temperature; PPT, precipitation; TAP, total annual precipitation; GS, growing season; GSL, growing season length; doy, day of year; d, days; SD, standard deviation

## Results

**The tundra-atmosphere carbon flux.** Net annual carbon loss from the alpine tundra ecosystem ranged from 177 g C m$^{-2}$ in 2010 to 314 g C m$^{-2}$ in 2013 and averaged 232 g C m$^{-2}$. The mean growing season length was 77 ± 7 days (mean ± 1 standard deviation) from 16 June to 27 August (Table 1); the standard deviation among methods used to calculate growing season length was 21 days (see Methods section for details)[39]. During the growing season, the mean cumulative gross primary productivity (GPP) was 118 g C m$^{-2}$ (Table 1), and a previous study using the Community Land Model (CLM; version 4.5) showed good agreement between the diurnal and seasonal cycles of measured and modeled GPP ($r = 0.86$ at daily resolution) at this location[40]. The mean cumulative growing season ecosystem respiration ($R_{eco}$; 64 g C m$^{-2}$; Table 1) was less than previously estimated from chamber (3 years; mean = 197 g C m$^{-2}$) and multiple linear regression (1 year; 130 g C m$^{-2}$; based on soil moisture and temperature) techniques[41], but prior results were based on less rigorously calculated and generally longer growing seasons. The mean cumulative growing season NEE was −41 g C m$^{-2}$, where negative values denote net carbon uptake by the surface.

Respiration fluxes dominated the NEE for approximately 9 months of the year, and the mean annual cumulative $R_{eco}$ was 341 g C m$^{-2}$. The $R_{eco}$ was most variable during the late winter months and the spring, and the magnitude of the carbon source during this time principally determined the ecosystem carbon balance on an annual basis (Fig. 2). The tundra functioned as a decreasing carbon sink during the growing season ($p = 0.07$; Fig. 3a) despite the absence of significant temporal trends in growing season GPP or $R_{eco}$. Net annual carbon emissions increased over the seven-year study period ($p = 0.07$; Fig. 3b), but neither the cumulative NEE nor the cumulative $R_{eco}$ were significantly correlated with seasonal air temperature or precipitation. The absence of clear environmental forcing highlights the potential influence of competing plant and microbial processes, temporal lags, and/or complex biogeophysical interactions on the trends identified by this work.

The mean annual carbon source strength of snow-scoured alpine tundra on Niwot Ridge (232 g C m$^{-2}$) was similar in magnitude to

the subalpine forest carbon sink (−217 g C m$^{-2}$)[9]. Although the majority of the western United States carbon sink is located in mountain regions[42,43], this result demonstrates that the areal subalpine/alpine ratio of a given area, together with the trajectory and variability of NEE from each respective ecosystem, ultimately determines the integrated ecosystem carbon sink or source status in mountainous terrain. That the alpine tundra was a stronger carbon source than has been previously estimated (175 g C m$^{-2}$)[9] was due to the relatively more robust gap-filling procedures used by this study, in combination with the increasing carbon source strength over time (most recent study only contained data through 2012). In the context of other alpine ecosystems around the world, the alpine tundra on Niwot Ridge was a greater source of carbon to the atmosphere than a Mediterranean alpine shrubland in Spain (annual mean NEE = 50 g C m$^{-2}$; 2 years)[7] and an alpine wetland meadow in China (annual mean NEE = 106 g C m$^{-2}$; 3 years)[8], neither of which was associated with permafrost. In this way, additional carbon respiration from permafrost degradation could factor into the carbon source strength difference between these ecosystems and the alpine tundra on Niwot Ridge.

**Soil carbon fractions and radiocarbon dynamics.** Tundra ecosystems are characteristically heterogeneous environments due to fine scale variations in topographic position and moisture status[29,34,41,44]. As a result, soil carbon pools and fluxes were measured and characterized with respect to radiocarbon content at four locations including one dry and moist meadow site, and two wet meadow sites on solifluction lobes that may be underlain by permafrost. Across the soil moisture gradient, the free light soil fraction (Lf; inter-aggregate particulate organic matter) ranged from 15.4 to 32.6% of the bulk soil on a per mass basis and was composed of 26.4 to 29.6% carbon. The radiocarbon content of the Lf soil fraction (148.1‰) was significantly higher at one of the wet meadow sites compared to any other site (10.6‰ to 66.2‰; Table 2). The combined occluded light and heavy soil fractions (Hf*; organic matter occluded within soil aggregates and/or complexed with minerals) composed the remainder of soil mass and contained between 4.9 and 9.4% carbon (Table 2). As is

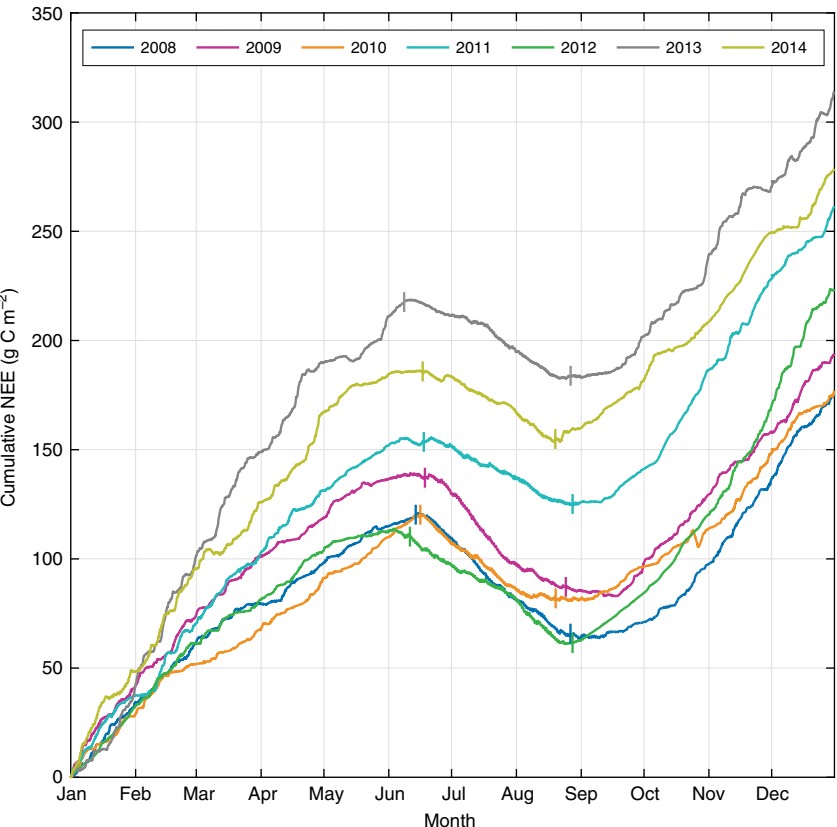

**Fig. 2** The alpine tundra was annually a net source of carbon to the atmosphere. The interannual variability of cumulative NEE of carbon dioxide between 2008 and 2014, shown as the equivalent grams of carbon per square meter. A positive slope denotes net carbon loss to the atmosphere and vertical dashes mark the beginning and end of each growing season determined as the average of three methodological techniques[39]

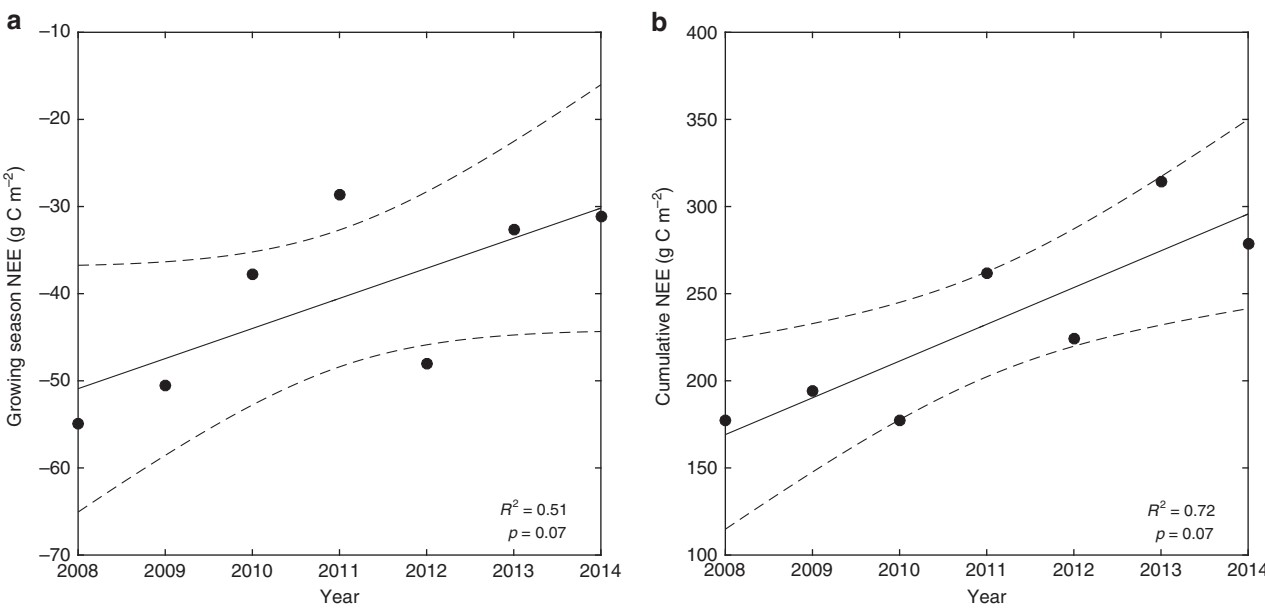

**Fig. 3** Carbon emissions from the alpine tundra ecosystem increased over time. Mann-Kendall regression analysis of the cumulative NEE of carbon dioxide during **a** the growing season and **b** the entire year between 2008 and 2014. Dashed lines correspond to the 95% confidence interval

typical, the Hf* fraction was radiocarbon depleted compared to the Lf and ranged from −66.2 to −98.7‰. During the summer, the mean $^{14}CO_2$ respired from all four sites ranged from 33‰ to 46‰. During the winter, gradient-method soil respiration[34] from the dry and moist meadow sites was undetectable and

therefore insufficient to prepare a sample for radiocarbon analysis. However, soil respiration continued throughout the winter at both wet meadow sites (mean winter respiration rate = 0.62 µmol m$^{-2}$ s$^{-1}$)[34], where the mean $^{14}CO_2$ of soil respiration was 33.8‰ and −16.9‰, respectively (Table 2).

**Table 2 Soil carbon pools at each sampling location and seasonal analysis of their respective radiocarbon signatures**

| Site | Season | Vegetation | Total C (kg m⁻²) | Lf (%) | Hf* (%) | Lf C (%) | Hf* C (%) | $\Delta^{14}$C CO$_2$ (‰) | $\Delta^{14}$C Lf (‰) | $\Delta^{14}$C Hf* (‰) |
|---|---|---|---|---|---|---|---|---|---|---|
| 13 | Summer | DM | 14.1 | 32.6 | 67.4 | 26.4 | 8.4 | 45.8 ($n = 5$) | 24.4 | −98.7 |
| 14 | Summer | MM | 10.8 | 16.4 | 83.6 | 26.9 | 9.7 | 44.7 ($n = 5$) | 10.6 | −86.5 |
| 17 | Summer | WM | 9.9 | 16.0 | 84.0 | 29.6 | 4.6 | 33.2 ($n = 8$) | 66.2 | −68.8 |
| 18 | Summer | WM | 10.6 | 15.4 | 84.6 | 27.2 | 9.6 | 36.0 ($n = 10$) | 148.1 | −72.5 |
| 17 | Winter | WM | 9.9 | 16.0 | 84.0 | 29.6 | 4.6 | 33.8 ($n = 5$) | 66.2 | −68.8 |
| 18 | Winter | WM | 10.6 | 15.4 | 84.6 | 27.2 | 9.6 | −16.9 ($n = 7$) | 148.1 | −72.5 |

Vegetation categories are dry meadow (DM), moist meadow (MM) and wet meadow (WM). Lf and Hf* represent the light and heavy (including occluded light) carbon fractions. Delta (Δ) notation denotes that samples have been $^{13}$C corrected. Site numbers correspond to a previous study at this location[41] and are bounded by the coordinates 40°03′07″ N; 105°35′11″ W and 40°03′09″ N; 105°35′14″ W

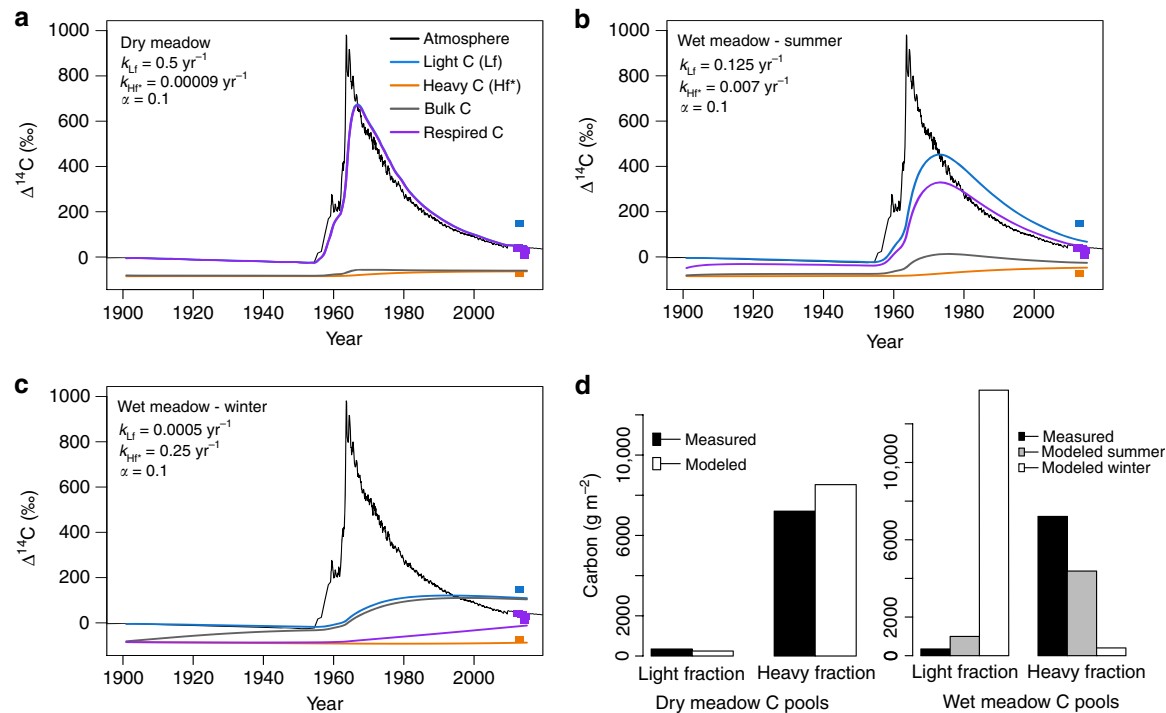

**Fig. 4** Radiocarbon modeling indicates wintertime respiration of older carbon from wet meadow control points on solifluction lobes. Model results from the **a** dry meadow, **b** wet meadow (summer), and **c** wet meadow (winter) scenarios show a lower decay rate of light fraction carbon ($k_{Lf}$) and a higher decay rate of heavy plus occluded light fraction carbon ($k_{Hf*}$) at the wet meadow site during winter. The input partitioning parameter ($\alpha$) describes the fraction of the light carbon pool that is transferred to the heavy carbon pool at each time step. **d** Seasonal comparison of modeled versus measured Lf and Hf* carbon fractions suggests non-steady-state carbon cycling at the wet meadow site, i.e., seasonal pools are not well captured

**Radiocarbon modeling.** Our results imply non-steady state carbon cycling with implications for the storage and release of older carbon. Winter respiration of Hf* carbon was specifically indicated by measurements from one wet meadow site that showed respiration of more depleted carbon during winter compared to summer (Table 2). These seasonal differences yielded modeled decay rates that were lower for Lf carbon ($k_{Lf}$) and higher for Hf* carbon ($k_{Hf*}$) during the winter relative to the drier sites or summer conditions at the same site (Fig. 4a–c). In contrast, the radiocarbon signature of respired carbon was always less depleted at the other wet meadow site (Table 2), and wintertime soil respiration was negligible at the dry and moist meadow sites, perhaps due to the lack of an insulating organic soil horizon[34,45].

Since the model runs at an annual resolution, the best-fit parameters determined from the seasonal flux data do not necessarily result in agreement between measured and modeled pool sizes, which reflect the long-term influence of the model conditions. In particular, the parameters required to fit winter respiration fluxes at the wet meadow site where respiration fluxes were depleted during winter resulted in very large discrepancies between the measured and modeled pool sizes (Fig. 4d). Accordingly, recent increases in the respiration Hf* carbon during the winter could be responsible for the relatively small Hf* pools at the wet sites compared to the dry sites (no wintertime soil respiration). From these results, it can be inferred that wintertime conditions at the wet meadow sites on solifluction lobes were uniquely conducive to microbial activity and that this scenario could be capable of stimulating decomposition of previously stable soil carbon.

**Potential contributing factors to an imbalanced carbon cycle.** Within the context of previous work at this site[31–33,36,37,46], increasing carbon emissions over time could be linked to respiration of a previously occluded carbon substrate that is augmenting present-day soil respiration from permafrost control points

throughout the snow-scoured tundra[34]. Since the NEE was not correlated with synoptic meteorology on an annual basis, the possibility of longer-term environmental changes and/or lags must be considered. Both summer air temperature (0.7 °C decade$^{-1}$) and annual positive degree days (DD; 157 DD decade$^{-1}$) have increased on Niwot Ridge since the middle of the twentieth century[37,38], and chronic nitrogen deposition has been shown to accelerate decomposition of intermediate-age SOM in dry alpine tundra[46]. Accordingly, the carbon cycle on Niwot Ridge may be responding to multi-decadal increases in air temperature and/or nutrient availability, which can stimulate respiration disproportionately more than primary productivity[47,48]. For example, carbon emissions are currently dominating over carbon uptake at high latitudes in response to increased mean annual air temperature of 0.35 °C decade$^{-1}$ since the 1970s[21]. Specifically, Alaskan heath, wet sedge, and tussock tundra were observed to be net annual sources of 20 g C m$^{-2}$, 84 g C m$^{-2}$, and 105 g C m$^{-2}$, respectively, to the atmosphere over recent 6–8 year periods[49,50]. The parallel trajectory of increasing net annual carbon loss identified by this work suggests that the alpine and arctic tundra carbon cycles may respond to the effects of environmental change in an analogous manner.

Disturbance can also affect carbon cycling in permafrost areas by changing the physical, hydrological, and/or biological characteristics of an ecosystem[17]. Solifluction, in particular, is one of the most widespread processes of soil movement in periglacial environments, and physical disturbance could be affecting the vertical distribution of soil organic carbon beneath solifluction lobes[20,51]. In this way, elevated $R_{eco}$ could result from ongoing solifluction that has exposed previously occluded carbon stocks, although this effect is not well documented in the literature[20]. Permafrost thaw can also result in soil drydown that promotes increased aerobic respiration[17], and hydrological reorganization can similarly affect vegetation with implications for nutrient allocation and productivity[10,40]. Previous work on the Qinghai-Tibetan Plateau hypothesized that slow vegetation recovery after hydrological disruption from permafrost collapse could result in large net annual carbon loss[10]; however, neither sudden hydrological nor vegetation changes have been observed at the long-term Niwot Ridge study plots[30,31], and increasing aeolian or dissolved organic or inorganic carbon outputs from the system would not affect the carbon fluxes reported here. Ultimately, it is uncertain which if any of these processes may be contributing to respiration of old carbon on Niwot Ridge, but sustained and potentially increasing net annual carbon emissions suggest that the alpine tundra ecosystem is not currently functioning at an equilibrium state.

This study may be the first to directly quantify the respiration of old soil organic matter from alpine tundra. Coupled to 7 years of continuous eddy covariance data, these results constrain the contribution of old carbon to currently observed patterns of mountain carbon cycling. Although snow-scoured areas represent only a small fraction of alpine tundra, these ecosystems are globally distributed and are generally representative of the locations where mountain permafrost occurs. As such, permafrost degradation could be progressively contributing to the carbon source strength of alpine tundra, thereby reducing the integrated carbon sink strength of mountain areas in general. Although the connections between high-latitude climate warming, permafrost degradation, and disturbance are increasingly recognized as important feedbacks to climate change, future eco-climatological projections generally neglect the role of other potentially sensitive systems in this scenario. These results imply that mountain ecosystems could contribute to this feedback in alpine areas where subsurface carbon stocks are vulnerable to environmental disturbance. Similarly, non-steady-state carbon cycling identified by this work could herald biogeochemical disequilibrium in other spatially heterogeneous, semiarid, or seasonally snow-covered ecosystems given the nature of alpine areas as sentinels of change.

## Methods

**Site description.** The snow-scoured alpine tundra study site was located on Niwot Ridge in the Colorado Rocky Mountains, USA[27]. Niwot Ridge begins on the Continental Divide and extends eastward as a narrow arête for about 2 km before it widens into a series of gentle slopes on rounded knobs down to the alpine treeline, approximately 8–9 km east of the Continental Divide[30,52]. In total, the Niwot Ridge alpine zone covers approximately 20 km$^2$ in area[31]. Over the period 1982 to 2012, the mean annual air temperature and precipitation near the study site were −2.2 °C and 884 mm (~75% in the form of snow), respectively, but snow cover was sparse due to windy conditions that scoured snow from the soil surface throughout the winter[27,53]. Continuous meteorological data have been collected on Niwot Ridge since the early 1950s, including the longest high-altitude air temperature record in the conterminous United States[54]. This makes Niwot Ridge an ideal place to study the effects of climate change and variability on the alpine tundra carbon cycle, and two identical 3-m eddy covariance (EC) towers, believed to be the highest in North America, were established near the T-Van site (40°03′06″ N; 105°35′07″ W; 3480 m asl) in 2007[26]. These towers measured the NEE of $CO_2$ over a temporally consistent statistical measurement footprint wherein 80% of the cumulative flux originated from within 400 horizontal meters predominantly to the west of each tower along the prevailing wind direction (Fig. 1a)[26,55]. At each tower, a co-located sonic anemometer (CSAT 3; Campbell Scientific, Logan, UT, USA) and infrared gas analyzer (LI-7500; LI-COR, Lincoln, NE, USA) were used to quantify the vertical wind fluctuations and the density of atmospheric $CO_2$, respectively[26,27]. The NEE was calculated as the covariance between instantaneous (10 Hz) deviations from the 30-min mean of the vertical wind speed and the scalar density of $CO_2$. Mann-Kendall and least squares regression analyses were used to identify temporal trends in the cumulative NEE, and statistical correlation between meteorology and the carbon cycle, respectively.

**Eddy covariance measurements.** Previous studies have focused extensively on the data quality from the T-Van study area including the potential for systematic uncertainty resulting from advective flows, sensor heating, and/or beneath-tower $CO_2$ storage, which is generally reduced due to the short measurement height, gently sloping terrain, prevailing windy conditions, and persistent westerly wind direction at the study site[26,27]. Post-processing of the EC data consisted of standard coordinate rotation and the Webb adjustment[56]. Missing values in the eddy covariance data were replaced by the average value under similar meteorological conditions[57], and the GPP and $R_{eco}$ were modeled following ref. [58]. We primarily used data from the west eddy covariance tower since it was located ~50 m closer to the solifluction lobes that may correspond to permafrost. However, during multi-day periods of consecutively missing west tower data (5.5% of all data), empirical between-tower relationships specific to daytime and nighttime growing season NEE, and NEE during the rest of the year, were used for infilling purposes. These significant relationships ($p \ll 0.001$) were developed using 1 year of simultaneous east and west tower data collection (data collected between November 2007 and October 2009), and validated using east tower measurements to predict the west tower data between 1 August 2013 and 31 December 2013. Using this approach, the modeled west tower cumulative NEE was 3.3% less than the observed west tower cumulative NEE during that time. There was no significant trend in the percentage of infilled values during the study period.

We used the average of three different methods to determine the growing season start and end dates and the variability around the resultant growing season length[39]. First, a carbon uptake period[59] was calculated in which the growing season length corresponds to the total number of days in which the 24-h mean NEE was less than zero. Second, 5-day and 10-day running averages of NEE were evaluated and the growing season start and end dates were identified as the days in which the NEE crossed from positive to negative values and vice versa[60]. Third, variable length regressions were calculated on 3-day to 9-day subsets of total daily NEE during the spring and fall. For this method, the day with the steepest negative slope during the spring defined the beginning of the growing season and the day with the shallowest positive slope in autumn defined the end of the growing season[61].

**Radiocarbon measurements.** Radiocarbon data are expressed as the parts per thousand (per mil; ‰) deviation from a standard (NBS Oxalic Acid I) of fixed isotopic composition:

$$\Delta^{14}C = [FM - 1]1000, \qquad (1)$$

where

$$FM = \frac{\left(\frac{^{14}C}{^{12}C}\right)sample}{\left(\frac{^{14}C}{^{12}C}\right)standard}, \qquad (2)$$

and FM is the fraction modern. To correct for mass-dependent fractionation, all samples were corrected to a common $\delta^{13}C$ value of −25‰ and also for radioactive decay between the time of sampling and measurement[62]. The $\delta^{13}C$ was measured

offline from $CO_2$ splits at the Institute of Arctic and Alpine Research (INSTAAR) Stable Isotope Laboratory at the University of Colorado Boulder[63].

To collect soil respiration samples for radiocarbon analysis during the growing season, semi-permanent 25-cm diameter × 10-cm deep soil collars[64] were buried to a depth of 5 cm at four alpine tundra sites including one dry meadow, one moist meadow, and two wet meadow locations on solifluction lobes. Soil respiration samples were collected by attaching a closed dynamic chamber system to the soil collar[65]; vegetation inside the soil collars was not clipped. Atmospheric (and soil) air from within the chamber was initially removed by circulating air (at 0.9 l min$^{-1}$; rate controlled by an adjustable flow meter) from within the chamber headspace through a desiccant ($Mg(ClO_4)_2$) and then through a column filled with soda lime. The $CO_2$ was scrubbed in this way until the equivalent of approximately three chamber volumes had passed over the soda lime, in order to maximize the removal of the atmospheric radiocarbon signature. Sample-containing air was then diverted around the soda lime trap to pass through a 2-l glass flask and a $CO_2$ analyzer (LI-820; LI-COR, Lincoln, NE, USA) before being exhausted back into the chamber headspace (flow rate was reduced to 0.5 l min$^{-1}$ during this step). The sample was isolated and collected by closing the valves on the flask when sufficient $CO_2$ had accumulated in the flask (~550 μmol mol$^{-1}$) to prepare a graphite target for radiocarbon analysis.

Due to the combination of persistent high winds and small magnitude respiratory fluxes, this chamber system was unable to capture sufficient carbon for radiocarbon analysis during the winter. As a result, two types of gas wells were deployed to each of the four sampling locations for wintertime radiocarbon analysis. The first type of gas well was constructed of a length of 0.64-cm diameter stainless steel tubing buried vertically in the soil to 15 cm depth (with ~10 cm of tubing extending above the soil surface to allow for sampling), then capped with a 0.64-cm Swagelok union fitted with a Teflon septum[66]. The second type of gas well was constructed of a 7-cm length of 3.8-cm diameter polyvinyl chloride (PVC) pipe capped with a two-hole #10 rubber stopper and sealed with silicone gel[34,67]. For sampling, two lengths of 0.64-cm diameter Teflon tubing extended from the interior of this gas well (through the two-hole rubber stopper) to a height of approximately 5 cm above the soil surface. Samples were collected from each of these gas wells by connecting a 470 cc pre-evacuated stainless-steel canister to the aboveground tubing (the second piece of tubing from PVC gas wells remained tightly capped) via a length of custom-crimped stainless-steel tubing that allowed the canister to fill over the period of approximately 1 h, in order to minimize disturbance to the soil $CO_2$ profile. The average difference between simultaneous measurements of $^{14}CO_2$ collected during the growing season via the two gas well methods was 0.5‰ ($n = 6$). In winter, the $^{14}CO_2$ collected from the stainless-steel gas wells was an average of 6‰ higher than the $^{14}CO_2$ from PVC gas wells at the wet meadow site 17, but 19‰ lower than the PVC gas wells at the other wet meadow location (site 18) ($n = 12$). All air samples were purified and graphitized by the INSTAAR Laboratory for AMS Radiocarbon Preparation and Research at the University of Colorado Boulder, then shipped to the Keck Carbon Cycle AMS Lab at the University of California, Irvine, for analysis.

Although atmospheric $CO_2$ was removed from the chambers prior to sampling, we accounted for leakage of atmospheric $CO_2$ into the chambers during the accumulation period (which is of particular importance when soil respiration fluxes are low) by calculating the amount of ambient $CO_2$ ($f_{air}$) in each sample based on the $\delta^{13}C$ ratios of $CO_2$ in the sample ($\delta^{13}C_{obs}$), ambient air ($\delta^{13}C_{air}$), and heterotrophic soil respiration ($\delta^{13}C_{resp}$):

$$f_{air} = \frac{\left(\delta^{13}C_{obs}\right) - \left(\delta^{13}C_{resp}\right)}{\left(\delta^{13}C_{air}\right) - \left(\delta^{13}C_{resp}\right)}, \quad (3)$$

where $\delta^{13}C_{air} = -8‰$ and $\delta^{13}C_{resp} = -28‰$[68]. Using this estimate of $f_{air}$, we then calculated air-corrected $^{14}C$ signatures ($\Delta^{14}C_{cor}$) from each observed soil respiration $^{14}C$ signature ($\Delta^{14}C_{obs}$):

$$\Delta^{14}C_{cor} = \frac{\Delta^{14}C_{obs} - (f_{air}\Delta^{14}C_{air})}{(1 - f_{air})}. \quad (4)$$

Since radiocarbon analysis of respired soil air yields only a bulk $\Delta^{14}C$ value, the SOM was separated by physical means prior to radiocarbon analysis. Soils from 5 to 15 cm depth at the dry, moist, and wet meadow sites were air dried, sieved through 2 mm mesh, and then fractionated into inter-aggregate particulate organic matter (free light fraction (fLf)), particulate organic matter occluded within soil aggregates (occluded light fraction (oLf)), and organic matter that is complexed with minerals (heavy fraction; Hf) fractions (based on a density of 1.85 g cm$^{-3}$) at the Lawrence Berkeley National Laboratory in Berkeley, California, USA[25]. All three soil fractions were similarly prepared for radiocarbon analysis at the INSTAAR Laboratory for AMS Radiocarbon Preparation and Research and shipped to the Keck Carbon Cycle AMS Lab for radiocarbon content analysis.

**Soil carbon model**. We used the software R to model the mean turnover time of the Lf (fLf) and Hf* (oLf + Hf) SOM fractions and their seasonal contribution to soil respiration from a dry meadow and two wet meadow sites on solifluction lobes. We chose these sites for modeling based on the availability of supporting information and contrasts in the respired $^{14}CO_2$ flux. For the wet meadow site, separate models were fit to the summer and winter respiration fluxes; winter respiration was

not detectable at the dry meadow site precluding fitting of a separate model. In total, we assessed the output from three different model parametrizations (dry meadow, wet meadow summer, and wet meadow winter). Differences between the parameter fits of the summer and winter models and their implications are addressed in the manuscript text.

We used the two-pool model in series from the SoilR package as our basic modeling framework, with the Lf and Hf* data from the density separation as constraints on the partitioning of the total carbon (TC) between the two model pools. Each model was initialized at time = 0 with a single depth increment and TC = 12.1 kg C m$^{-2}$, which was measured from depth-distributed sampling of the wet meadow site. We then ran the non-steady state models with annual time steps for the period from 1901 to 2015. Based on ref. [69], we applied estimated annual carbon inputs of 72 and 124 g C m$^{-2}$ for dry and wet meadow simulations, respectively. We derived the radiocarbon content of annual inputs from an interpolation of pre- (IntCal13) and post-bomb (Northern Hemisphere zone 2)[70] atmospheric values, without a time-lag.

Each of the models was initialized with values for model fitting parameters including the first-order decay constants of the light ($k_{Lf}$) and heavy ($k_{Hf*}$) pools as well as an input partitioning parameter ($\alpha$), which describes the fraction of the light pool that is transferred to the heavy pool at each time step. Initial values for carbon pool rate constants were set by assuming an equal contribution from each pool to the observed mean annual soil flux estimates and the initial value was arbitrarily set to $\alpha = 0.25$. We evaluated the parameter combinations resulting in the best model fit to the data using the FME package in R. Specifically, we varied the model fitting parameters, and the model output from each parameterization was compared to observations using a cost function routine. The model cost function considered both the agreement between measured and modeled values of the $^{14}CO_2$ of soil fluxes and the $^{14}C$ content of the Lf and Hf* pools. Using this cost function routine, the parameter space was searched using a pseudo-random parameter selection to narrow the best-fit parameters at a coarse scale. A Levenberg–Marquardt algorithm method was subsequently implemented to further constrain the best-fit parameter combination, using the results of the pseudo-random method as the input. For each model fitting routine, the same minimum and maximum values for the parameter search space were used. After completion of the model fitting procedure, we ran the model using the best-fit parameter combination. It is important to note that these model best-fit parameters are representative of the instantaneous decay rates required to best match the observed seasonal respiration fluxes and do not necessarily equate to the true long-term decay rates of the model pools.

## Data availability

The daily eddy covariance data are available at https://doi.org/10.6073/pasta/10fb65e51cd04631bb80c82288b5c51a; 30-min and 10-Hz data are available from the authors upon request. The radiocarbon data are available at https://doi.org/10.6073/pasta/467c6ed086546bc968c568bc1bfea5f9.

## Code availability

The eddy covariance processing and radiocarbon modeling scripts are available from the authors upon request.

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

## Acknowledgements

This study was funded by NSF awards BCS 1129562 to J.F.K. and DEB 0423662, DEB 1027341, and DEB 1637686 to the Niwot Ridge LTER. We thank Scott Lehman and Dave Bowling for improving our radiocarbon sampling design, Cristina Castanha for help with soil density fractionations, and Natascha Kljun and Carlos Sierra for supporting the Flux Footprint Prediction and radiocarbon modeling analyses, respectively. William Kolby Smith provided valuable feedback on an early version of this manuscript. Publication of this article was funded in part by the University of Colorado Boulder Libraries Open Access Fund. Any use of trade, firm, or product names is for descriptive purposes only and does not imply endorsement by the U.S. Government.

## Author contributions

J.F.K., P.D.B., and M.W.W. designed the study, J.F.K. collected the data, J.F.K. and C.R.L. analyzed the data, J.F.K. wrote the manuscript, and J.F.K., P.D.B., C.R.L., and M.W.W. contributed to the final manuscript version.

## Additional information

**Competing interests:** The authors declare no competing interests.

