## [Peer Review File · Nature Communications]

Reviewers' comments:

Reviewer #1 (Remarks to the Author):

This paper entitled "Increasing carbon loss from snow-scoured alpine tundra: An indicator of climate change?" reports decreasing carbon uptake during the growing season and increasing carbon release throughout the year at the Niwot Ridge site in the western US. The authors performed a relatively long-term observational study of net ecosystem exchange and performed an age partitioning using ^{14}C methods. The paper shows a striking and persistent change in NEE, with a strong shift towards net carbon release to the atmosphere. The authors discuss several interesting patterns, including the relative importance of alpine tundra and alpine forest, since most of the carbon sink in the western US comes from mountainous regions. I think this paper is an important contribution to our understanding of how alpine systems respond to climate change. I outline a few concerns below about the representativeness of the sites, the limits of the age modeling, and the title and overall flow of the manuscript.

1. The authors present this work as representative of the whole alpine tundra biome (e.g., lines 38), but it is unclear how much of the biome is similar to these sites, which are snow-scoured and experiencing solifluction. Solifluction can strongly influence soil carbon profiles, primary productivity, and net ecosystem carbon balance. Could some of the high carbon release and decrease of sink strength be due to soil disturbance rather than climate change? The lack of a correlation with annual climate conditions (e.g., line 137) could suggest that a more general ecosystem disturbance mechanism is responsible for the changes. More fundamentally, it was not clear to me how representative these sites were of the Niwot ridge research area specifically and alpine tundra generally. The authors invoke the importance of this small-scale heterogeneity, and it would strengthen the manuscript if the results were interpreted in this framework. It would be useful to have a map of the site showing the extent of the flux tower footprint and major vegetation or topographic features.

2. The authors appear to use sound methods to calculate the modern fraction of soil organic matter, but these results are not well integrated into the text. The per-mil values are reported, but the modeling results are only briefly touched upon, except in the final figure. Additionally, it was not very clear if these data were from a single site (the wet meadow) or multiple sites throughout the research area. The radiocarbon data would be much easier to interpret and more impactful if they were better interpreted and integrated into the text.

3. The introduction of the paper seems somewhat imbalanced—claiming that little to nothing is known about alpine tundra NEE and that the high-latitude permafrost carbon feedback is well constrained. I think the paper generally overstates the lack of understanding of alpine tundra (e.g. lines 26, 45, 73) and overstates the understanding of the overall permafrost carbon feedback. In reality, there have been many carbon-balance studies in both alpine (lots of literature from the Niwot, the Canadian Rockies, and the Tibetan Plateau) and high-latitude cold regions. Nevertheless, the net response of these areas remains highly uncertain because of the multiple competing

responses acting on different timescales. These complicating dynamics include plant-soil thermal interactions, effects of hydrological reorganization, and non-vertical ecosystem carbon fluxes (Jorgenson & Osterkamp 2005; Lawrence et al. 2015; Abbott et al. 2016; Mu et al. 2017). I think that highlighting some of these permafrost complexities in the introduction and final section of the paper would strengthen the overall story. Likewise, simplifying the title (removing the phrase after the colon) would make the paper's message clearer.

Abbott, B.W., Jones, J.B., Schuur, E.A.G., III, F.S.C., Bowden, W.B., Bret-Harte, M.S., et al. (2016). Biomass offsets little or none of permafrost carbon release from soils, streams, and wildfire: an expert assessment. *Environ. Res. Lett.*, 11, 034014.

Jorgenson, M.T. & Osterkamp, T.E. (2005). Response of boreal ecosystems to varying modes of permafrost degradation. *Can. J. For. Res.*, 35, 2100–2111.

Lawrence, D.M., Koven, C.D., Swenson, S.C., Riley, W.J. & Slater, A.G. (2015). Permafrost thaw and resulting soil moisture changes regulate projected high-latitude CO₂ and CH₄ emissions. *Environ. Res. Lett.*, 10, 094011.

Mu, C.C., Abbott, B.W., Zhao, Q., Su, H., Wang, S.F., Wu, Q.B., et al. (2017). Permafrost collapse shifts alpine tundra to a carbon source but reduces N₂O and CH₄ release on the northern Qinghai-Tibetan Plateau. *Geophys. Res. Lett.*, 2017GL074338.

Reviewer #2 (Remarks to the Author):

Please see attached Review file.

Reviewer #3 (Remarks to the Author):

To Knowls et al.

General comments:

The authors made a great effort on investigating the carbon budget at the snow-scoured alpine meadow on a high mountain by combining the 7 years eddy covariance flux and soil decomposition modeling based on radiocarbon analysis. The story was well written to state that this ecosystem has been a net source of carbon recently, which could be caused by the newly started decomposition of permafrost-associated carbon. The context of this study well suits to the target of Nature Communications.

However, there is few unclear points for better understanding of carbon dynamics of alpine meadow ecosystem.

Figure 2 showed the increasing trend in NEEs both in growing season and whole year though there was a problem that the relatively high p ($=0.07$) was not convincing us much. However, there were no clear trends in onset, offset and length of growing season and air temperature and precipitation during 7 measurement years. On the other hand, the authors found that there would have been long-term increasing trends in both summer air temperature and annual positive degree days from mid 20th century to present, and tried to say that those warmings were the potential causes for enhanced carbon decomposition resulting in recent net carbon loss. However again, p ($= 0.09$) of trend line was not so small, and also there was no guarantee that the net carbon loss has existed only for recent period and/or has been increasing gradually together with warming trend.

About the increasing summer NEE, I like to know about which was the stronger cause: decreasing GPP or increasing Reco? At least GPP is to be shown in this MS for growing season to clarify it, though this approach would introduce another systematic error in daytime GPP estimation due to potential error in daytime Reco estimation based on the nighttime NEE-Tsoil relationship that usually contains large uncertainty.

From ^{14}C radiocarbon analysis, the authors speculated that net carbon loss was largely from the addition of newly started decomposition of Hf under the warming climate at wet meadow. That story is really interesting. However, the authors analyzed only a snap shot data taken for one time, and the comparison between wet and dry meadows, and winter and summer seasons.

I am not sure if they are enough to talk about the long-term trend.

Simply, I am just worried if the increasing trend did not exist, and if there is any other cause to result in the net carbon loss. For ex., there would be other carbon outgoing flux (for ex., DOC/DIC fluxes via soil water movement, soil carbon removal by wind, etc.) and be a systematic error in eddy measurement due to coldness, low air pressure and slope inclination in alpine condition.

Overall, the story was really good and the use of both eddy and radiocarbon data was very unique. It could be worth for re-considering of the publication after the revision.

Reviewers' comments:

Reviewer #1 (Remarks to the Author):

This paper entitled "Increasing carbon loss from snow-scoured alpine tundra: An indicator of climate change?" reports decreasing carbon uptake during the growing season and increasing carbon release throughout the year at the Niwot Ridge site in the western US. The authors performed a relatively long-term observational study of net ecosystem exchange and performed an age partitioning using ^{14}C methods. The paper shows a striking and persistent change in NEE, with a strong shift towards net carbon release to the atmosphere. The authors discuss several interesting patterns, including the relative importance of alpine tundra and alpine forest, since most of the carbon sink in the western US comes from mountainous regions. I think this paper is an important contribution to our understanding of how alpine systems respond to climate change. I outline a few concerns below about the representativeness of the sites, the limits of the age modeling, and the title and overall flow of the manuscript.

Thanks to Reviewer 1 for a very constructive review.

1. The authors present this work as representative of the whole alpine tundra biome (e.g., lines 38), but it is unclear how much of the biome is similar to these sites, which are snow-scoured and experiencing solifluction... More fundamentally, it was not clear to me how representative these sites were of the Niwot ridge research area specifically and alpine tundra generally. The authors invoke the importance of this small-scale heterogeneity, and it would strengthen the manuscript if the results were interpreted in this framework. It would be useful to have a map of the site showing the extent of the flux tower footprint and major vegetation or topographic features.

We have elevated (from the methods section) and expanded upon a portion of the previous site description to address the representativeness of snow-scoured tundra in the introduction section (L68-84). Alpine tundra is a fundamentally windy environment and snow-scoured zones are thus common to alpine areas, but we now clearly state that “these locations represent only a fraction of the heterogeneous alpine environment” (L74-75). This is further characterized in the context of previous work that has called these areas “the most active geomorphic, hydrologic, and biologic zones in the area” (L75-76). We also include additional description of the formation and location of solifluction lobes on Niwot Ridge and a new figure (Figure 1) that illustrates the eddy covariance tower footprint with respect to the heterogeneity of vegetation and topography.

Solifluction can strongly influence soil carbon profiles, primary productivity, and net ecosystem carbon balance. Could some of the high carbon release and decrease of sink strength be due to soil disturbance rather than climate change? The lack of a correlation with annual climate conditions (e.g., line 137) could suggest that a more general ecosystem disturbance mechanism is responsible for the changes.

We now discuss solifluction as a potential contributor to observed net annual carbon loss as part of more general revisions away from climate change attribution (L205-211). Instead, the root cause of our results is presented as an open question that includes environmental change, disturbance, or some combination thereof (L184-222; L231-236; more details below in our response to your General Comment 3).

2. The authors appear to use sound methods to calculate the modern fraction of soil organic matter, but these results are not well integrated into the text. The per-mil values are reported, but the modeling results are only briefly touched upon, except in the final figure. Additionally, it was not very clear if these data were from a single site (the wet meadow) or multiple sites throughout the research area. The radiocarbon data would be much easier to interpret and more impactful if they were better interpreted and integrated into the text.

We now specify that “soil carbon pools and fluxes were measured and characterized with respect to radiocarbon content at four locations including one dry and moist meadow site and two wet meadow sites on solifluction lobes that may be underlain by permafrost” (L142-145). The per-mil values are reported in Table 2 and discussed on L148-159, and the radiocarbon modeling section has been extensively revised with a focus on more effectively weaving the site-specific decay rates and their implications through the text (L162-182).

3. The introduction of the paper seems somewhat imbalanced—claiming that little to nothing is known about alpine tundra NEE and that the high-latitude permafrost carbon feedback is well constrained. I think the paper generally overstates the lack of understanding of alpine tundra (e.g. lines 26, 45, 73) and overstates the understanding of the overall permafrost carbon feedback. In

reality, there have been many carbon-balance studies in both alpine (lots of literature from the Niwot, the Canadian Rockies, and the Tibetan Plateau) and high-latitude cold regions. Nevertheless, the net response of these areas remains highly uncertain because of the multiple competing responses acting on different timescales. These complicating dynamics include plant-soil thermal interactions, effects of hydrological reorganization, and non-vertical ecosystem carbon fluxes (Jorgenson & Osterkamp 2005; Lawrence et al. 2015; Abbott et al. 2016; Mu et al. 2017). I think that highlighting some of these permafrost complexities in the introduction and final section of the paper would strengthen the overall story. Likewise, simplifying the title (removing the phrase after the colon) would make the paper's message clearer.

We used this comment to guide a general reinterpretation of our results toward the integrated response to a set of potentially complex ecosystem processes that may or may not align temporally or involve feedbacks (e.g. L47-51; L87-89; L92-95; L184-222; L231-236). The reviewer is correct that it's not certain this is a simple climate change story and the revised manuscript more clearly differentiates what is known (the tundra is a significant and potentially increasing net annual source of carbon dioxide; old carbon respired from permafrost sites contributes to this flux during the winter) from what is unknown (this is likely due to some interaction between climate, soil, hydrological, and/or ecological processes but we can't be sure of exactly which ones). These results both improve our understanding of complex ecosystem dynamics in alpine areas (more in the following paragraph) and highlight the need for ongoing research in permafrost systems. We've retitled the manuscript "Evidence for non-steady-state carbon emissions from snow-scoured alpine tundra" since both the radiocarbon and eddy covariance datasets suggest disequilibrium, and this subject will be of broad interest to the ecological and climatological research communities.

Although there have been many carbon balance studies in high-altitude and high-latitude cold regions, eddy covariance data are sparse during the winter, and we are not aware of any eddy covariance-based studies from sites associated with mountain permafrost. When coupled to the radiocarbon model, the resulting experimental design is capable of quantifying net annual carbon emissions from this system and also the seasonal carbon sources that contribute to this flux, all within the context of the longest high-altitude climate record in the conterminous United States. We believe these results significantly improve current understanding of the magnitude and trajectory of high altitude carbon cycling.

Abbott, B.W., Jones, J.B., Schuur, E.A.G., III, F.S.C., Bowden, W.B., Bret-Harte, M.S., et al. (2016). Biomass offsets little or none of permafrost carbon release from soils, streams, and wildfire: an expert assessment. *Environ. Res. Lett.*, 11, 034014.

Jorgenson, M.T. & Osterkamp, T.E. (2005). Response of boreal ecosystems to varying modes of permafrost degradation. *Can. J. For. Res.*, 35, 2100–2111.

Lawrence, D.M., Koven, C.D., Swenson, S.C., Riley, W.J. & Slater, A.G. (2015). Permafrost thaw and resulting soil moisture changes regulate projected high-latitude CO₂ and CH₄ emissions. *Environ. Res. Lett.*, 10, 094011.

Mu, C.C., Abbott, B.W., Zhao, Q., Su, H., Wang, S.F., Wu, Q.B., et al. (2017). Permafrost collapse shifts alpine tundra to a carbon source but reduces N₂O and CH₄ release on the northern Qinghai-Tibetan Plateau. *Geophys. Res. Lett.*, 2017GL074338.

We cite these papers as evidence that our results may be affected by complex and possibly competing ecosystem responses to climatological and/or physical disturbance.

Reviewer #2 (Remarks to the Author):

Overarching comments:

This ms is positioned within an important domain of geo- and climate sciences: large-scale carbon-climate feedback. This is a general area that attracts large interest as it may affect future climate and it holds substantial uncertainties.

The current study provides a novel perspective on the carbon exchange of an alpine meadow over 7 years. While this study is limited to one site it does contribute useful insights into the operation of such a system.

I find this study to be well designed and executed. Its scope would make it a strong contribution to a leading journal in its field. There are several reasons while I find it less fitting for a journal like Nature Communications. The basic one is that it is not clear that permafrost carbon-climate feedback in alpine tundra may at all be considered a potentially significant part of the earth system w.r.t. positive feedback to climate warming. Key considerations are detailed below.

We appreciate this perspective and thank the reviewer for the detailed review.

On the key rationale for the study to be considered in Nature Communications: the potential magnitude of carbon-climate feedback from thawing alpine permafrost: The basic premise of this study is that this process in this earth component may give a significant positive feedback to warming climate.

The paper cites correctly an estimate that mountain permafrost may hold about 66 Pg C (4.5% of global soil C). This may be contrasted to the roughly half of the global soil carbon stock that is held in just the top 1-3 m of circum-Arctic tundra and taiga permafrost (Topsoil/Surface-PF; ~1000 Pg-C) with deeper layers below as Deep-PF (~400 Pg-C) and in Pleistocene Ice Complex Deposit permafrost (ICD-PF, a.k.a. Yedoma; ~400 Pg-C) (e.g., Hugelius et al., 2014).

Unfortunately, none of the earth system – climate models analyzed for IPCC AR5 in 2013 accounted for any permafrost carbon emissions; a requirement for the next IPCC report to not miss this climate feedback. Recently, however, one-dimensional earth system models are providing a first approximation of the scale of the feedback from terrestrial permafrost thawing. Model scenarios forecast an average carbon release from gradual deepening of the surface active layer of permafrost by year 2100 under current climate warming trajectory (Representative Concentration Pathway RCP8.5) of about 100 Pg C (Schuur et al., 2015).

This level of circum-Arctic permafrost carbon release is of the same scale as land-use change, but nearly a factor of ten lower than from current direct anthropogenic CO₂ emissions, and would add an estimated 0.1-0.3°C global warming by year 2100 (IPCC, 2013; Schuur et al., 2015). The Arctic “amplification” of climate warming is accounted for in this. Even if we assume that mountain permafrost is subjected to similar forcing enhancement as in the Arctic, a scaling estimate would suggest that mountain permafrost carbon-climate feedback hence would be at maximum on the order of 0.01-0.03 °C over the coming hundred years.

This is not resolvable. This undermines the central motivation articulated for the broad importance of the current study.

This is a valid and logically presented critique. In response, we highlight several novel contributions of this work: (1) Regardless of magnitude, no study has empirically demonstrated sustained carbon emissions or respiration of old carbon from alpine tundra. Consequently, a paradigm of disequilibrium between alpine tundra ecosystems and the environment is new information that will be relevant to researchers across the geosciences and life sciences disciplines. (2) Greater than expected (from previous data collected at non-permafrost alpine sites) carbon loss from alpine tundra mitigates the carbon sink strength of mountain ecosystems that account for most of the carbon sink in semi-arid areas including the western United States. Revising this number down to account for significant alpine tundra carbon emissions would affect the regional carbon balance in these areas. (3) This type of work promotes novel comparison between high latitude and high-altitude systems where similarities or differences reveal important information about these systems, but also about the function of other spatially heterogeneous, seasonally snow-covered, herbaceous, or other ecosystems as well. This raises the question of what other ecosystems might be expected to behave in this way now or in the future? In this way, we expect that a more comprehensive understanding of the complex feedbacks highlighted in this work could be relevant to a range of global ecosystem types.

Other considerations:

Temporal aspect: Seven years, while being a massive field effort, is still a short observational period on which to draw conclusions of temporal trends related to changing climate forcing. The increasing carbon loss trend (Figure 3) is no longer presented as central to our main argument. Instead, the manuscript is now focused on (a) sustained net annual carbon emissions in excess of other non-permafrost alpine tundra ecosystems, and (b) radiocarbon measurements that demonstrate respiration of older carbon during the winter. Both of these results suggest non-steady-state ecosystem function, which is how we've re-framed the principal conclusion (and the title) of this work. In parallel, we've softened our attribution to climate change in favor of an open-ended discussion that includes other disturbances (e.g. L184-222).

This was a massive field effort and we appreciate the recognition. It was no small task to maintain data collection during the winter, and we suspect that this represents the longest continuous (most alpine studies are growing season only) alpine eddy covariance dataset in the world.

Spatial aspect: While the LTER at Niwot Ridge is well positioned and important, it may be best suitable for process-oriented studies. It seems risky to use results from this meadow to infer on carbon fluxes for alpine permafrost globally.

This is a fair point that may be alleviated somewhat by our no longer invoking climate warming as the root cause of non-steady-state carbon cycling observations. Nevertheless, the LTER site was selected to generally represent high-altitude tundra in this region and uniquely offers the opportunity to contextualize our results within the longest high-altitude climate (and associated vegetation) record in the conterminous United States.

Reviewer #3 (Remarks to the Author):

To Knowles et al.

General comments:

The authors made a great effort on investigating the carbon budget at the snow-scoured alpine meadow on a high mountain by combining the 7 years eddy covariance flux and soil decomposition modeling based on radiocarbon analysis. The story was well written to state that this ecosystem has been a net source of carbon recently, which could be caused by the newly started decomposition of permafrost-associated carbon. The context of this study well suits to the target of Nature Communications. However, there is few unclear points for better understanding of carbon dynamics of alpine meadow ecosystem.

Thanks to Reviewer 3 for the thoughtful review.

Figure 2 showed the increasing trend in NEEs both in growing season and whole year though there was a problem that the relatively high p ($=0.07$) was not convincing us much. However, there were no clear trends in onset, offset and length of growing season and air temperature and precipitation during 7 measurement years. On the other hand, the authors found that there would have been long-term increasing trends in both summer air temperature and annual positive degree days from mid 20th century to present, and tried to say that those warmings were the potential causes for enhanced carbon decomposition resulting in recent net carbon loss. However again, p ($= 0.09$) of trend line was not so small, and also there was no guarantee that the net carbon loss has existed only for recent period and/or has been increasing gradually together with warming trend.

You're correct to point out that the past and/or future duration of our observations is not well constrained. Hence, the physical underpinnings of our results are now presented as an open question that includes potential interactions between climate change and disturbance (e.g. L184-222; see response to Reviewer 1). The revised paper has been subsequently refocused to more strictly highlight the data-based conclusions i.e. that independent eddy covariance and radiocarbon analyses suggest non-steady-state function with respect to carbon cycling. The impacts of this result may not be exclusive to alpine tundra (see response to Reviewer 2), and we hope that our work will spur further mechanistic inquiry into this broadly relevant question.

About the increasing summer NEE, I like to know about which was the stronger cause: decreasing GPP or increasing Reco? At least GPP is to be shown in this MS for growing season to clarify it, though this approach would introduce another systematic error in daytime GPP estimation due to potential error in daytime Reco estimation based on the nighttime NEE-Tsoil relationship that usually contains large uncertainty.

Table 1 has been expanded to include cumulative sums of both GPP and R_{eco} during the growing season, and we now state that the "tundra functioned as a decreasing carbon sink during the growing season ($p = 0.07$; Figure 3a) despite the absence of significant temporal trends in growing season GPP or R_{eco} " (L117-119). Although modeling the GPP and R_{eco} fluxes from NEE (following Reichstein et al. 2005) does introduce the potential for systematic error as you note, the ecosystem carbon source strength during the fall, winter, and spring principally determined the carbon balance for the year (L113-117).

From ^{14}C radiocarbon analysis, the authors speculated that net carbon loss was largely from the addition of newly started decomposition of Hf under the warming climate at wet meadow. That

story is really interesting. However, the authors analyzed only a snap shot data taken for one time, and the comparison between wet and dry meadows, and winter and summer seasons. I am not sure if they are enough to talk about the long-term trend.

Multiple radiocarbon measurements were collected over a period of years at all sites, but we agree that these data only represent a snapshot of the long-term conditions. Consequently, the revised manuscript places greater emphasis on the general disequilibrium between carbon losses and gains relative to the potential for trends.

Simply, I am just worried if the increasing trend did not exist, and if there is any other cause to result in the net carbon loss. For ex., there would be other carbon outgoing flux (for ex., DOC/DIC fluxes via soil water movement, soil carbon removal by wind, etc.) and be a systematic error in eddy measurement due to coldness, low air pressure and slope inclination in alpine condition.

A new discussion paragraph addresses the possibility for other permafrost-associated disturbances such as solifluction, hydrological reorganization, and/or vegetation change to affect the observed carbon fluxes, and the entire manuscript has been restructured to reflect this modified perspective. We now specify that “increasing aeolian or dissolved organic or inorganic carbon outputs from the system would not affect the carbon fluxes reported here” (L217-219) since they are not measured by eddy covariance instrumentation. Similarly, we now include a specific data quality statement: “previous studies have focused extensively on the data quality from the T-Van study area including the potential for systematic uncertainty resulting from advective flows, sensor heating, and/or beneath-tower CO₂ storage, which is generally reduced due to the short measurement height, gently sloping terrain, prevailing windy conditions, and persistently westerly wind direction at the study site (Blanken et al. 2009; Knowles et al. 2012)” (L266-270). Indeed, the current study builds on a decade-long foundation of chamber-, tower-, and model-based research at this site and in many ways represents the culmination of this work.

Overall, the story was really good and the use of both eddy and radiocarbon data was very unique. It could be worth for re-considering of the publication after the revision.

Thank you!

Reviewers' comments:

Reviewer #1 (Remarks to the Author):

This paper entitled "Evidence for non-steady-state carbon emissions from snow-scoured alpine tundra" reports decreasing carbon uptake during the growing season and increasing carbon release throughout the year at the Niwot Ridge site in the western US. The authors performed a relatively long-term observational study of net ecosystem exchange using flux tower methods and performed an age partitioning using ^{14}C methods. The paper shows a striking and persistent change in NEE, with a strong shift towards net carbon release to the atmosphere. The authors discuss several interesting patterns, including the relative importance of alpine tundra and alpine forest, since most of the carbon sink in the western US comes from mountainous regions. I think this paper is an important contribution to our understanding of how alpine systems respond to climate change. The authors have carefully revised the paper in response to the three previous reviewers and I consider the paper now ready for publication. Just a few line edits below:

1. Line 39: thawed instead of degraded permafrost might be clearer/easier to understand
2. Line 76: hot spots is the standard spelling, though many papers have mistakenly spelled this like the wifi connection "hotspot" (Berhardt et al 2017)
3. Line 188: same as above (hot spot)
- 4.

Bernhardt, Emily S., Joanna R. Blaszczak, Cari D. Ficken, Megan L. Fork, Kendra E. Kaiser, and Erin C. Seybold. "Control Points in Ecosystems: Moving Beyond the Hot Spot Hot Moment Concept." *Ecosystems* 20, no. 4 (June 2017): 665–82. <https://doi.org/10.1007/s10021-016-0103-y>.

Reviewer #4 (Remarks to the Author):

This is an overall interesting study about the carbon balance of the one alpine site Niwot Ridge. The authors make use of a 7-year long record of eddy-covariance based NEE observations in combination with ^{14}C analysis and modelling of soil respiration. While I can see the great work behind the study and its potential in general, I am sorry that unfortunately I cannot be very positive about the broad relevance of the study and the robustness of the conclusions.

1) Relevance to the audience of Nature Communications.

This study seems to be perfect for the audience of topical journals in the field of biogeochemistry. The results can be interesting for people working with alpine permafrost regions. Unfortunately, I cannot see the results being interesting for a broad scientific audience because

- still understanding of the causes of year-to-year changes in observed NEE still remains unclear and speculative
- there is no judgment why these alpine ecosystems should matter at all for global carbon-climate feedback mechanisms. The overall amount of soil carbon is low, area is low and I do not see a potential high change in fluxes in future that could matter for the overall carbon balance.

2) Overall Comments.

The overall hypothesis is that long-term climate change led to higher heterotrophic soil respiration (probably from previously permanently frozen ground). However, I cannot see unfortunately the data proving or rejecting this hypothesis. Most of the discussion is rather speculation: Given the data presented, it COULD BE that increasing soil respiration from warming soil is the reason. The warm year 2012 shows the opposite: Lower annual NEE probably due to higher GPP, which shows

the importance of plant process relations to environmental factors (but why does the curve follows 2008 in spring?) Still, lag effects in soil processes COULD play a role for high NEE estimates in 2013 and 2014. In general, 7 data points and the presented p-value are not valid for stronger conclusions.

Eddy-covariance based NEE results do have their own uncertainty, in particular in such location, e.g. due to gap filling, advection, footprint dynamics. Such uncertainties need to be thoroughly included into the overall data analysis. It remains unclear if there are consistent changes of the footprint or gaps over time. But even more important: NEE always represents a mixture of GPP, autotrophic respiration and heterotrophic respiration. Therefore, for a robust conclusion about a recent change in alpine permafrost soil heterotrophic respiration and its relevance on the global scale I would expect a combined analysis of observations of many sites with process-based models.

Reviewers' comments:

Reviewer #1 (Remarks to the Author):

This paper entitled "Evidence for non-steady-state carbon emissions from snow-scoured alpine tundra" reports decreasing carbon uptake during the growing season and increasing carbon release throughout the year at the Niwot Ridge site in the western US. The authors performed a relatively long-term observational study of net ecosystem exchange using flux tower methods and performed an age partitioning using ^{14}C methods. The paper shows a striking and persistent change in NEE, with a strong shift towards net carbon release to the atmosphere. The authors discuss several interesting patterns, including the relative importance of alpine tundra and alpine forest, since most of the carbon sink in the western US comes from mountainous regions. I think this paper is an important contribution to our understanding of how alpine systems respond to climate change. The authors have carefully revised the paper in response to the three previous reviewers and I consider the paper now ready for publication. Just a few line edits below:

Thanks to Reviewer 1 for their valuable feedback throughout the review process and the recommendation to publish the manuscript.

1. Line 39: thawed instead of degraded permafrost might be clearer/easier to understand
Changed to "thawed" (L39).

2. Line 76: hot spots is the standard spelling, though many papers have mistakenly spelled this like the wifi connection “hotspot” (Bernhardt et al 2017)

3. Line 188: same as above (hot spot)

“Hotspots” has been changed to “control points” throughout the text following the Bernhardt reference (L78; L192; L772).

4. Bernhardt, Emily S., Joanna R. Blaszczak, Cari D. Ficken, Megan L. Fork, Kendra E. Kaiser, and Erin C. Seybold. “Control Points in Ecosystems: Moving Beyond the Hot Spot Hot Moment Concept.” *Ecosystems* 20, no. 4: 665–82. <https://doi.org/10.1007/s10021-016-0103-y>.

We now cite this paper.

Reviewer #4 (Remarks to the Author):

This is an overall interesting study about the carbon balance of the one alpine site Niwot Ridge. The authors make use of a 7-year long record of eddy-covariance based NEE observations in combination with ¹⁴C analysis and modelling of soil respiration. While I can see the great work behind the study and its potential in general, I am sorry that unfortunately I cannot be very positive about the broad relevance of the study and the robustness of the conclusions.

We thank Reviewer 4 for this perspective.

1) Relevance to the audience of *Nature Communications*.

This study seems to be perfect for the audience of topical journals in the field of biogeochemistry. The results can be interesting for people working with alpine permafrost regions. Unfortunately, I cannot see the results being interesting for a broad scientific audience because

We note that the current study promotes novel comparison between high latitude and high-altitude systems where similarities or differences not only reveal important information about these areas, but also about the function of other lower elevation, spatially heterogeneous, seasonally snow-covered, or semi-arid ecosystems as well (L35-36; L240-243). In this way, we expect that a more comprehensive understanding of the complex feedbacks highlighted by this work will be relevant to many global ecosystem types, and believe that the revised manuscript now aligns with the *Nature Communications* aims and scope to publish papers that “represent important advances of significance to specialists within each field”.

- still understanding of the causes of year-to-year changes in observed NEE still remains unclear and speculative

We agree that our data are not sufficient to quantitatively interpret the causes of year-to-year variability in observed NEE. However, that interpretation does not factor into the main conclusion of our study i.e., that persistent net annual carbon loss coupled to winter respiration of old carbon substrate supports a non-steady state paradigm for a sensitive, widespread, and understudied ecosystem type.

- there is no judgment why these alpine ecosystems should matter at all for global carbon-climate

feedback mechanisms. The overall amount of soil carbon is low, area is low and I do not see a potential high change in fluxes in future that could matter for the overall carbon balance. Although the magnitude of the mountain permafrost feedback to warming may not be a major direct contributor to global scale environmental change (this is currently unknown), the potential effects of this feedback are not directly proportional to global carbon stocks. No study has empirically demonstrated sustained carbon emissions or respiration of old carbon from alpine tundra. As a result, a paradigm of biogeochemical disequilibrium between alpine tundra and the environment is new information that will be relevant across a broad range of environmental science disciplines including vegetation dynamics, nutrient cycling, hydrological processes, surface-atmosphere coupling, and/or the ensuing impacts on ecosystem services:

“By the end of the century, [the] mountain cryosphere will have changed to an extent that will impact the landscape, the hydrological regimes, the water resources, and the infrastructure. The impacts will not remain confined to the mountain area but also affect the downstream lowlands, entailing a wide range of socioeconomical consequences... Whereas an upward shift of the tree line and expansion of vegetation can be expected into current periglacial areas, the disappearance of permafrost at lower altitudes and its warming at higher elevations will likely result in mass movements and process chains beyond historical experience” (Beniston et al. 2018 with respect to European mountains, but applicable to all mid-latitude mountain areas).

Further, evidence of greater-than-expected alpine tundra carbon loss characterized by this work (compared to previous data collected at non-permafrost alpine sites) has the potential to mitigate the carbon sink strength of mountain ecosystems that account for the majority of carbon uptake in semi-arid areas including the western United States. Revising this number down to account for higher alpine tundra carbon emissions would significantly affect the regional carbon balance in these areas (L35-36; L128-132; L233-235).

Our revised manuscript includes new citations to Beniston et al. 2018 (additional broader impacts, cross-system links, and their socio-economic consequences), Williams et al. 2002 (sensitive mountain ecosystems as early warning indicators of more widespread changes), and Seastedt et al. 2004 (mountains as sources of water and nutrients to lower elevations) to further convey the broad implications of non-steady-state carbon cycling in the mountains.

Beniston, M. et al. The European mountain cryosphere: a review of its current state, trends, and future challenges. *Cryosphere* **12**, 759–794 (2018).

Williams, M.W., Losleben, M.V. & Hamann, H.B. Alpine areas in the Colorado Front Range as monitors of climate change and ecosystem response. *Geogr. Rev.* **92**, 180–191 (2002).

Seastedt, T.R. et al. The landscape continuum: a model for high-elevation ecosystems. *BioScience* **54**, 111–121 (2004).

2) Overall Comments.

The overall hypothesis is that long-term climate change led to higher heterotrophic soil respiration (probably from previously permanently frozen ground). However, I cannot see

unfortunately the data proving or rejecting this hypothesis. Most of the discussion is rather speculation: Given the data presented, it COULD BE that increasing soil respiration from warming soil is the reason. The warm year 2012 shows the opposite: Lower annual NEE probably due to higher GPP, which shows the importance of plant process relations to environmental factors (but why does the curve follows 2008 in spring?) Still, lag effects in soil processes COULD play a role for high NEE estimates in 2013 and 2014. In general, 7 data points and the presented p-value are not valid for stronger conclusions.

Our overall hypothesis does not invoke climate change: “We specifically set out to (1) characterize the net annual carbon source strength of a high-altitude alpine tundra site, and to (2) evaluate the relative contributions of distinct soil carbon pools to the observed ecosystem respiration fluxes on a seasonal basis” (L90-93). A previous manuscript draft did seek to establish causal links to long-term climate change, but this attribution was replaced with an open-ended discussion that includes the potential for other processes to affect our results (see our previous response to Reviewer 1). We have taken care in the revised manuscript to ensure that our main conclusions are strictly fact-based i.e., we focus on (a) sustained net annual carbon emissions in excess of other non-permafrost alpine tundra ecosystems, and (b) radiocarbon measurements that demonstrate respiration of older carbon during the winter. Both of these results suggest non-steady-state ecosystem function, which is how we have re-framed the principal contribution (and the title) of this work.

We generally agree that Figure 3 is “not valid for stronger conclusions” given the (potentially competing) influences of soil, plant, and meteorological processes on the data as well as the possibility of lag effects as you accurately note. As a result, the physical underpinnings of our results are presented as an open question that includes the potential for interactions between climate, soil, vegetation, and disturbance (e.g. L188-226). Although we do believe that the significant trends shown by Figure 3 are noteworthy and warrant inclusion in the manuscript, we have added text to specifically convey that they could result from a multiplicity of potentially competing factors, lags, and/or complex interactions (L122-125).

Eddy-covariance based NEE results do have their own uncertainty, in particular in such location, e.g. due to gap filling, advection, footprint dynamics. Such uncertainties need to be thoroughly included into the overall data analysis. It remains unclear if there are consistent changes of the footprint or gaps over time. But even more important: NEE always represents a mixture of GPP, autotrophic respiration and heterotrophic respiration. Therefore, for a robust conclusion about a recent change in alpine permafrost soil heterotrophic respiration and its relevance on the global scale I would expect a combined analysis of observations of many sites with process-based models.

Previous studies have quantified the random and systematic measurement uncertainty associated with the T-Van eddy covariance system including the statistical measurement footprint, energy balance closure, and the potential for horizontal advection, sensor heating, and beneath-tower carbon dioxide storage (Blanken et al. 2009; Knowles et al. 2012). In fact, the current study builds on a decade-long foundation of chamber-, tower-, and model-based research at this site and in many ways represents the culmination of this work. During this time, we observed no significant changes in the turbulent fluxes or the wind vector, land cover, or the percentage of

gapfilled data, which is now specified in the text (L220-221; L261-264; L291-292). Additional information about the eddy covariance data quality control and the gapfilling protocol can be found on L273-292.

To our knowledge, the T-Van site on Niwot Ridge is one of only two unmanaged alpine tundra eddy covariance sites in the world, (the other being Scholz et al. 2018), and the only such site where radiocarbon data has been collected. Hence, a formal analysis of observations from many alpine tundra sites is not possible. However, the revised manuscript now makes reference to Scholz et al. 2018 as well as a new study (Treat et al. 2018) that supports spatial heterogeneity as a common attribute of tundra ecosystems (L145-146).

With respect to models, a previous study (Wieder et al. 2017) did couple the T-Van eddy covariance data to the National Center for Atmospheric Research (NCAR) Community Land Model (CLM); good agreement between the measured and modeled sensible and latent heat fluxes and *GPP* provides a supporting line of evidence for the results of the current study (L105-108). We acknowledge that modeling the *GPP* and respiration fluxes from *NEE* does introduce the potential for systematic error, but we note that the ecosystem carbon source strength during the fall, winter, and spring (when vegetation was senesced and *GPP* did not contribute to *NEE*) principally determined the carbon balance for the year (L114-117).

Scholz, K., Hammerle, A., Hiltbrunner, E. & Wohlfahrt, G. Analyzing the effects of growing season length on the net ecosystem production of an alpine grassland using model-data fusion. *Ecosystems* **21**, 982–999 (2018).

Treat, C.C. et al. Tundra landscape heterogeneity, not interannual variability, controls the decadal regional carbon balance in the Western Russian Arctic. *Glob. Change Biol.* **24**, 5188–5204 (2018).

REVIEWERS' COMMENTS:

Reviewer #4 (Remarks to the Author):

Dear authors,

Thank you for clarifying all the issues that I had with that manuscript before and for further improving this manuscript. I support the publication of this study.